# 'It's all about patient safety': an ethnographic study of how pharmacy staff construct medicines safety in the context of polypharmacy

Nina Fudge, Deborah Swinglehurst

Institute of Population Health Sciences, Barts and The London School of Medicine and Dentistry, Queen Mary University of London, London, UK

**Correspondence to**
Professor Deborah Swinglehurst; d.swinglehurst@qmul.ac.uk

## ABSTRACT

**Objective** As polypharmacy increases, so does the complexity of prescribing, dispensing and consuming medicines. Medication safety is typically framed as the avoidance of harm, achievable through adherence to policies, guidelines and operational standards. Automation, robotics and technologies are positioned as key players in the elimination of medication error in the face of escalating demand, despite limited research illuminating how these innovations are taken up, used and adapted in practice. We explore how 'safety' is constructed and accomplished in community pharmacies in the context of polypharmacy.

**Design and setting** In-depth ethnographic case study across four community pharmacies in England (December 2017–July 2019). Data collection entailed 140 hours participant observation and 19 in-depth interviews. Practice theory informed the analysis.

**Participants** 33 pharmacy staff (counter staff, technicians, dispensers, pharmacists).

**Results** In their working practices related to polypharmacy, staff used the term 'safety' in explanations of why and how they were doing things in particular ways. We present three interlinked analytic themes within an overarching narrative of care: caring for the technology; caring for each other; and caring for the patient. Our study revealed a paradox: polypharmacy was visible, pervasive and productive of numerous routines, but rarely discussed as a safety concern per se. Safety meant ensuring medicines were dispensed as prescribed, and correcting errors pertaining to individual drugs through the clinical check. Pharmacy staff did not actively challenge polypharmacy, even when the volume of medicines dispensed might indicate 'high risk' polypharmacy, locating the responsibility for polypharmacy with prescribing clinicians.

**Conclusion** 'Safety' in the performance of practices relating to polypharmacy was not a fixed, defined notion, but an ongoing, collaborative accomplishment, emerging within an organisational narrative of 'care'. Despite meticulous attention to 'safety', carefully guarded professional boundaries meant that addressing polypharmacy per se in the context of community pharmacy was beyond reach.

## Strengths and limitations of this study

► Adopts an ethnographic approach, observing the practices of pharmacy staff 'doing safety' in the particular context in which it happens, rather than relying on interview accounts.
► Two researchers conducted observations, interviews and analysis, allowing different professional perspectives to inform the analysis, enhancing the study's credibility.
► Some aspects of the setting were inaccessible, for example, management decisions about 'running the business' of a community pharmacy.
► Our findings may not translate readily outside the context of independent community pharmacies, but our interpretations offer useful ways of conceptualising safety across UK and international settings.

## INTRODUCTION

Safety in prescribing, dispensing and administering medicines is a global public health priority.[1–5] WHO defines medicines safety as 'protect(ing) patients from harm while maximising the benefits from medication'.[6] Efforts to address medicines safety have traditionally targeted secondary care. Attention is now turning to primary care settings where a growing population of older people with multimorbidity, escalating prescribing and polypharmacy drive a greater burden of iatrogenic harm.[6–10] Community pharmacies provide a key role within medicines management in primary care, integrating multiple tasks related to medication (including but not limited to dispensing and delivering prescriptions, dispensing medicines through multi-compartment compliance aids (MCCAs) and counselling patients through medication use reviews (MURs)) with the need to operate as viable commercial businesses.[11] This work has become increasingly challenging and complex due to polypharmacy, but how safety is enacted and produced in community

pharmacy settings amidst this complexity remains poorly understood.[12 13]

Polypharmacy is usually defined as the concurrent use of five or more medications.[6 14] Differences in the way polypharmacy is defined and medication data is collected make international comparisons challenging, but several international studies demonstrate increasing polypharmacy in older populations.[15] The risks of polypharmacy are well documented: medicine errors, adverse drug reactions, falls, frailty, hospital admission, increased hospital stay and death.[16 17] In recognition that in some cases people may need multiple item prescriptions to treat their conditions, a distinction has been made between 'appropriate' polypharmacy or 'problematic' polypharmacy.[18]

Medication safety is typically framed as avoidance of error. Professional regulators and leadership bodies encourage reporting of errors, metrics for safe prescribing[5] and removal of inherent weaknesses in the system.[6] A more nuanced approach is now emerging, shifting the focus away from how errors are produced and avoided (which draws attention to a minority of incidents) towards what can be learnt from observing ordinary everyday performance, where—mostly—practice occurs error-free (a 'Safety II' approach).[19]

Previous work on safety in community pharmacies has presented safety culture as a product of managing the complex relationship between medicines-related workloads and maintaining commercial viability.[20] While safety improvement may be enabled by managers and institutional policies, it is enacted by front-line staff and therefore evolves as a result of this enactment.[11] One study, based on interview data, has suggested that implementation of procedures is subject to an interplay between procedures-as-imagined and procedures-as-done.[21 22] Adopting the term 'organisational resilience', Thomas et al[21] call for further research to investigate how organisational culture contributes to decision making and action around implementation of standardised procedures.

We are adopting a safety II orientation to our study of polypharmacy,[23 24] conducting ethnographic and participatory research across primary care settings (patients' homes, general practice, community pharmacy), seeking to generate 'practice-based evidence'.[25] We address the research question: How is patient safety constructed in community pharmacy settings in the context of polypharmacy?

## METHODS
### Setting
This study was part of an in-depth, multisite ethnographic study of polypharmacy in primary care (APOLLO-MM: *Addressing the Polypharmacy Challenge in Older People with Multimorbidity* - protocol previously published).[23] We conducted an organisational ethnography of routines and practices in four English community pharmacies, pseudonymised Willow, Foxglove (part of Woodland Independent Pharmacy Group), Poppy and Lilac (part of Meadow

| Table 1 | Characteristics of study pharmacies | | | |
|---|---|---|---|---|
| Location | Index of multiple deprivation 2019* | Study pharmacy | No of staff | |
| Woodland Independent Pharmacy Group (3 pharmacies) Urban setting 6% of local population 65 years or older (55% female, 47% male)† | 30% most deprived LSOA‡ | Willow | 2 pharmacists 1 trainee pharmacist 8 dispensers/ technicians 2 counter staff 1 delivery driver | Group pharmacist owners (n=2) who oversee pharmacies and cover staff when needed. |
| | 50% most deprived LSOA | Foxglove | 2 pharmacists 2 counter staff 8 dispensers/ technicians | |
| Meadow Independent Pharmacy Group (5 pharmacies) Suburban setting 17% of local population 65 years or older (55% female, 45% male)† | 20% least deprived LSOA | Poppy | 1 pharmacist 2 dispensers 1 part-time counter staff 1 delivery driver | Group pharmacist owner (n=1) who oversees pharmacies and covers staff when needed. |
| | 50% least deprived LSOA | Lilac | 1 pharmacist 1 dispenser 2 dispensers/counter staff 1 delivery driver | |

*https://www.gov.uk/guidance/english-indices-of-deprivation-2019-mapping-resources#indices-of-deprivation-2019-explorer-postcode-mapper.
†https://fingertips.phe.org.uk/profile/health-profiles.
‡LSOA or neighbourhood.
LSOA, layer super output area.

Independent Pharmacy Group) (see table 1). Pharmacies were recruited along with general practitioner (GP) practices to form contrasting research clusters. Each cluster (GP + pharmacies) provide medicine services to our patient participants in the wider APOLLO-MM study.[23]

Two researchers (NF, a social anthropologist and DS, an academic GP, both experienced ethnographers) undertook data collection and analysis. Data included: ethnographic observations; shadowing staff, inviting them to 'talk me through what you are doing'; formal interviews; documents (eg, standard operating procedures (SOPs), dosette checklists, to do lists, manufacturer's guidance identified as relevant through our observations and interviews). We conducted 140 hours observation (December 2017–January 2018; December 2018–February 2019, March–July 2019), focusing on everyday routines and practices concerning polypharmacy and the management of patients with multimorbidity. This work occurred primarily in 'backstage' regions of the pharmacies,[26] such as dispensaries and areas designated for preparation of MCCAs—known as 'dosettes' at all sites.

We used one-to-one briefing sessions, posters and information leaflets to ensure informed consent from participants. We adopted a 'processual consent' approach, revisiting consent iteratively before each occasion of observation.[27]

We conducted 19 formal interviews with 21 pharmacy staff identified through ethnographic observations as doing work relevant to polypharmacy and its safety (including one group interview) (see table 2). Interviewees signed consent forms in advance. We adopted a narrative approach, using a broad topic guide, inviting in-depth accounts of working practices (online supplemental file 1). We asked participants to attend the interview prepared to share a story relating to polypharmacy. Interviews lasted 14–59 min, were audio-recorded and transcribed verbatim.

### Data analysis
The dataset comprised 280 pages of typed fieldnotes, 279 pages of transcribed interviews and 46 documents. Fieldnotes were typed after observations/interviews (usually within 24 hours), incorporating reflections and theoretical insights, then shared between NF and DS to prompt further critical reflection guided by our different fieldwork experiences and disciplinary orientations. This informed subsequent fieldwork, and ensured a coherent approach to data gathering. We kept a digital, reflective journal using Evernote, sharing memos, observations and theoretical insights relevant to the wider polypharmacy project. We used QSR NVivo V.12 qualitative data analysis software for data management.[28] The data we present are anonymised; names for pharmacies, interviewees and staff are pseudonyms.

The analysis was inductive, with 'safety' emerging early in our analysis as an organising frame for the work of repeat dispensing for patients experiencing polypharmacy. We directed our analysis to the tasks and routines staff undertook, how these constituted notions of patient safety, and how staff accomplished safety-in-practice.[19] Informed by practice theory, we focused on interconnections between people, artefacts, spaces and technologies. Under this lens, organisations are conceptualised as 'bundles of practices' and management is an activity aimed at ensuring that social and material activities work more-or-less in alignment.[29 30]

### Patient and public involvement
We have a project advisory group of 11 members (academics, health professionals, representation from Age UK, two patient members, lay chair). An online patient panel of five members were involved in: proposal development, design of participant materials and project website (www.polypharmacy.org.uk), application for ethical approval, project launch event, piloting of interviews, study design and conduct.

### RESULTS
'Safety' was a collective concern in all pharmacies. Staff used the term 'safety' in explanations of how and why they did certain things: picking medicines from shelves in a particular order; asking patients for names and addresses; switching tasks regularly. In naturalistic talk, staff did not articulate what constitutes 'safety', but in interviews they referred to safety as 'the right drug, the right patient, the right time'.

The pharmacies used various technologies (eg, robots, MCCAs, spreadsheets, computers, telephones, printouts, labels, post-it notes) in the process of dispensing (see table 3). Willow has two robots which automate in-house dispensing and MCCA production, acting as a 'dosette production hub' for all the Woodland pharmacies. Poppy

| Table 2 | Pharmacy interviews | | | | | |
|---|---|---|---|---|---|---|
| Pharmacy | Pharmacist/pharmacy group owner | Pharmacist | Pharmacy technician | Dispenser | Counter staff | Total |
| Willow | 1 (m=1) | 3 (f=1, m=2) | 1 (m=1) | 4 (f=3, m=1) | | 9 |
| Foxglove | | 2 (m=2) | 2 (m=1) | 3 (f=2, m=1) | | 7 |
| Poppy | 1 (f=1) | | 1 (m=1) | 1 (f=1) | | 3 |
| Lilac | | | | 1 (f=1) | 1 (f=1) | 2 |

f, female; m, male.

**Table 3** Detailed ethnographic description, (known as 'thick description') of the four community pharmacies[47]

**Woodland independent pharmacy group**

| | |
|---|---|
| Willow pharmacy, housed in purpose-built premises, has a small footprint given the volume of medicines it dispenses. Much of their work comes from the GP surgery over the road. Every bit of space is taken up with stores of medicines: small boxes of pills in blister packs on shelves from counter to ceiling; large canisters of barrier creams, syrups and fortified milkshakes on lower shelves below the counters; robot-prepared dosette boxes stacked in floor-to-ceiling shelves. Delivery men visit twice daily, trolleys laden with large cardboard boxes full of medicines that are unpacked, cross-checked off the order sheet and stacked on the shelves as quickly as they are removed. At regular intervals, the low hubbub of dispensing and dosette box production work is punctuated by a woman at the front counter calling out '*Prescription waiting!*' as she clips the prescription on a tiny metal hanger at the front of the dispensary. The low whooshing noise from the dispensing robot signals drugs ready for a technician and pharmacist to check, bag up and pass to a customer. All 10–15 staff move quickly around the space as they gather up medicines, stack the shelves, deblister tablets to replenish the hungry robots, add a reminder to the whiteboard or post-it note, respond to customers—switching seamlessly between languages as needed. They skillfully navigate the tight space, shifting boxes, climbing steps, passing medicines from one to the other and always listening out for each other. | Foxglove, sister pharmacy of Willow, has an even smaller footprint than Willow. It's housed in a converted Victorian building, within a row of shops with flats above. Entering the pharmacy through an automatic sliding door signals the shop has recently been modernised. A padded bench along one wall allows customers to sit while they wait. A selection of over-the-counter medicines is on display. The front counter is staffed by two people, handling patients who hand over their paper prescriptions or ask to pick up medicines. Counter staff flick through a card file to find a customer's prescription which cross-checks to a numbered shelf where medicines have been bagged up, waiting for patients to collect them. The dispensary is at the back with an eye-level counter, giving staff cover as well as a view of what's happening on the shop floor. The dosette area is hidden from customer view, even further back in this Tardis-like building and is particularly narrow. Here, a counter runs the length of one wall, which the 'dosette team' use to check the robot-produced dosettes (sent over from Willow) for errors before they are given to patients. Pharmacists tip and flick the dosette box from underneath to check and count the capsules and tablets in each cell. Floor to ceiling shelving runs around the room, storing each patient's four-weekly supply of dosettes, stored alphabetically by patient's surname and according to their collection or delivery day. As with Willow, there are on average ten staff diligently working away, but always with an ear out to help one another. |

**Meadow independent pharmacy group**

| | |
|---|---|
| A steep slope marks the entrance to Poppy pharmacy, located opposite a GP practice, in a quiet, residential part of a suburban town. I wonder how some older customers navigate this entrance, but a sign on the door tells people to ask for help if needed. Each time the door opens a tune bleeps out, signalling the arrival or departure of a customer or delivery man. This immediately prompts someone to leave the dispensary which is in a raised area at the back of the shop, to leave their tasks, come forward and ask '*how can I help?*' The shop floor houses an array of over-the-counter medicines and beauty items. It has a welcoming feel, with a row of chairs opposite the counter for customers waiting for prescriptions or just needing a seat. There are usually two or three people working in the shop at a time, in quiet dedication to their tasks. When there are no customers, staff focus on dispensing prescriptions, preparing baskets for filling dosettes, checking dispensed medicines, answering the phone, ordering medicines and receiving deliveries. Every available wall space is full of shelving to house medicines. Sticky, fluorescent yellow labels stuck to the shelves remind staff to '*select with care*'. | Lilac pharmacy is in a parade of shops: a builder's trade shop, fish and chip shop, Co-op mini supermarket and a café in a suburban residential area. I am struck by a large sign plastered along the length of the front window 'FREE DELIVERY', similar to a sign on the glass shop front at Poppy. Inside it's very calm and quiet—the door opens onto a spacious, airy and light shop with shelves displaying all manner of over-the-counter medicines and beauty products, even children's toys. People frequently come in for a chat with the counter staff—often without even the excuse of a prescription to pick up. Despite the large shop floor, space behind the counter in the dispensary and dosette areas is tight—when I'm there, as with all the other pharmacies, I feel I am in the way although nobody seems to mind. I notice all the worktops are black, and Leena, the pharmacy manager, explains that all the pharmacies in the group are replacing their worktops with black ones. I am told that black worktops make the white tablets easier to see: when checking and counting tablets staff can simply lay the clear plastic dosette box on the worktop and easily see the number of pills in each cell. Compared with the Woodland group pharmacies, pace of work at the Meadow group pharmacies is less frenetic. |

GP, general practitioner.

and Lilac had considered automating dosette production but reported it was not financially viable.

Preparation, storage and delivery of dosettes to support medicines-taking was a prominent working routine across both pharmacy groups, with demand growing despite reports questioning their safety.[31][32] Meadow Group's manager reported '*we are taking on extra work*' since larger, chain pharmacies were reducing this service. A hand-written poster at Willow showed dosette production increasing 10% from 252 to 278 patients over 7 months. All pharmacies used electronic prescribing systems (EPS) to receive prescriptions from GPs and order prescriptions from GPs on patients' behalf.

We present three interlinked themes which show how staff accomplish safety within an organisational narrative of care: caring for the technology; caring for each other; caring for the patient.

## Caring for the technology

Two technologies, dosette robots and EPS, illustrate how new routines and discourses emerge when technologies are introduced, and how 'traditional' technologies such

as paper charts, post-it notes and whiteboards remain critical to the safety of medicines routines.

Staff at Woodland pharmacies regularly described the dosette robot, and their work with it, as contributing to patient safety: '*Number one reason is patient safety*'; '*Everything is for patient safety*'; '*We got the robot for patient safety*'. This well-rehearsed collective narrative appealed to staff and drove the implementation and ongoing use of the robot, although staff were never explicit about what constitutes '*safety*' nor how the robot contributed to it. We interpret staffs' statements about safety as resonating with the robot manufacturer's such as '*increased accuracy*' compared with the '*manual preparation method*' enabling '*the pharmacy to greatly increase safety*' (Document: robot manufacturer's website). The website also boasts that dosettes are '*proven to boost adherence rates from 61% to 97%*', and that the robot offers '*competitive advantage*' through '*lower production costs*', '*increase in production speed*', and a '*significant decrease in labour costs*' (Document: robot manufacturer's website). The presumed economic benefits were invisible to us; staff did not express any connection between robot and revenue.

Automation of dosette production has not eliminated the need for human care. The robot is never left alone when '*in production'* (ie, filling dosettes). It has generated new scope for errors and new working routines to address errors. For example, staff must constantly '*replenish'* the robot's 400 translucent blue containers; drugs in underfilled containers are beyond the reach of the robot's suction arm which lifts drugs one-by-one into the programmed dosette cells. This requires technical knowledge about a drug's stability and sophisticated local knowledge about how fast the lines are—'*slow lines*' demand less frequent attention than '*fast lines*'. The '*replenishing*' process involves staff '*deblistering*' tablets or capsules from their packaging by feeding blister packs through a deblistering machine, depositing tablets into shallow plastic trays for labelling by drug name, batch number, description and number of tablets. A coworker pours the tablets into the matched robot container, first selecting the correct container lid—with holes just big enough to enable the suction mechanism to pick out a single tablet for each cell. After filling, staff do manual checks to ensure the dosette contains the right number of items per cell and that pills have not accidently '*jumped*' from cell to cell as sometimes happens. Certain medicines ('*externals*') are not kept in the robot; they may have limited shelf life after deblistering or be '*slow lines'.* Staff add these manually once the robot has completed a patient's 4-week set.

The pharmacy has protocols and standard operating procedures but when we enquire of their whereabouts the manager says they are '*locked in a cupboard in the staff room.'* Probing further we discover that although the robot was installed over 2 years ago they are '*still working*' on their protocol for dosette production which '*changes all the time*'. The pharmacy manager continues '*we are so busy and demand is increasing. When we reach the pinnacle we will write*

*something. I usually just ask the staff and they tell me what's the best way*' (Fieldnote, Willow, 09/01/2018, DS). His quote illustrates the importance of local knowledge, grounded in practice, and how this becomes constructed through informal communication rather than formal protocol.

EPS is central to dispensing routines, but pharmacy staff adopted a range of workarounds to ensure safety that developers of the EPS system may not have envisaged. It was striking that across all pharmacies all prescriptions sent electronically by GPs (via the National Health Service (NHS) 'spine') were printed. The working day was punctuated at c.15 minute intervals by staff '*refreshing*' the EPS screen to view and print new scripts, a routine they considered essential to safe working practice. Only paper had sufficient 'ecological flexibility'[33] to enable swift movement of the prescription around different physical spaces in ways which supported their collaborative working—the same paper prescription passed from person-to-person moving in and out of different working routines. Its materiality was important, serving as a handy checklist for picking medicines off the shelves, checking medicines before '*bagging up'*, and handing the '*right medicines to the right patients*' (see table 4). Jimi, Woodland's manager, explained that portable electronic devices would not do the job; they have no access to a secure wireless internet connection. Given his innovation with robot technologies, we interpret this not as resistance to technology per se, rather an expression that the technology is insufficiently flexible for tasks required.[33 34]

## Caring for each other
Caring for each other encompasses a number of practices to ensure safe working as a team. Members of staff were always alert, always listening and ready to take initiative. Staff were able to report errors without fear of negative consequences and we experienced a strong commitment to social cohesion.

Throughout our fieldwork we observed staff diligently pursuing their individual tasks but always poised and available to help a colleague, for example, find a drug, check/sign-off a prescription, move to the dispensary to serve a growing queue of patients, replenish the robot:

Caleb, a technician, arrived at work and mumbled something. Rohima turned to him and said '*she's gonna have to call back in half an hour for dosette box issues*'. I realised I had no idea which question she was responding to, or where it had come from. There is such a strong sense in this workplace of people being on the alert to subtle cues from colleagues around the site and I realise I am not always 'tuned in' and wonder at what seems like an 'extra sense' operating between its members to keep things flowing.

(Fieldnote, Willow, 03/01/2018, DS)

We often witnessed staff huddled together to solve a problem, with junior staff encouraged to offer solutions, define processes or suggest procedural changes:

**Table 4** Electronic prescribing routines

| Routines involving the printed copy of the electronic prescription | Examples from fieldnotes and interviews |
|---|---|
| Refreshing the screen, printing out newly arrived prescriptions, leafing through the printouts to organise the workflows and routines:<br>▲ Does everything look in order?<br>▲ Any obvious anomalies?<br>▲ Anything urgent?<br>▲ Anything need to be ordered for the next day? | When he's not serving at the counter, Mahendra, the pharmacist, is at the dispensary computer dealing with prescriptions. He has refreshed the screen and printed out a batch of prescriptions. As he processes each prescription on the computer, he puts the paper prescription in a red plastic basket with labels which he's also printed out: a bag label and a label for each medicine prescribed. At this point he may also order any medicines needed. He stacks prepared baskets on top of each other. This pile of baskets waits for a dispenser to take a basket, pick and select the required drugs, before handing back to Mahendra to do the final check and bag up the medicines.<br>(Fieldnote, Poppy, 25/03/2019, NF)<br><br>With a pile of freshly printed out prescriptions, Sameer, the pharmacist, explains that you can't just sort them by patient and put them in the basket for the dispensers to start processing, because there might be urgent ones that need to be processed straight away. Instead, you *'have to sift through to see what is important.'* After visually scanning each prescription, he has identified three prescriptions which need processing first. One is for a rescue pack for a COPD patient; an acute situation as the patient will want the drugs today. Another acute case—a urine infection. The third one is for meal replacement drinks. These will need to be ordered today so the patient can get them tomorrow. Sameer continues with his explanation *'so I put these ones (the thicker pile of the recently printed prescriptions) behind this lot (a pile of prescriptions already in the grey basket) because they came through last and then I put these three (the urgent ones) on top because they are important.'*<br>(Fieldnote, Willow, 09/01/2019, NF)<br><br>Shabnam is now going through the pile of recently printed electronic prescriptions. She is sorting through them at speed and putting certain ones to one side. She explains that she *'tries to take out the bigger ones, leave the easier ones for the others to do 'cos I'm more experienced'* and they can do the easier ones in between dealing with *'waitings or collections'*.<br>(Fieldnote, Willow, 10/01/2019, NF) |
| Paper prescriptions can easily move about the pharmacy and be transferred from one member of staff to another to complete a dispensing routine | Mo took a small yellow card marked 'Willow pharmacy' at the top of which was a sticker with a patient's name and address details. He looked in the black filing cabinet under the relevant patient's surname, pulled out a prescription then put a box of multivitamin tablets and the prescription into a small, red, plastic basket. He went to the front counter and I heard him explain that *'there is only one'*. He then returned to the black filing cabinet, found another green prescription marked *'repeat dispensing'* and made up another small red basket containing the script and a box of emollient, and took this to the front counter. Linda took over, taking hold of the basket. Mo moved quickly round the corner to dispense some methadone in the supervision corner. I realise I am once again a bit perplexed about this apparently simple transaction—and am yet to fully grasp the path of prescriptions around this front counter area. The problem is that it is so fast moving and the routines are so tacit and collectively embodied that nothing much is said and opportunity to ask them to explain is limited by the frenetic nature of the environment. I wonder how Nina is managing with this as clearly we do need to grasp the path of a prescription request from the front counter to the dispensing of medicines. Mo had disappeared and I had lost my 'shadowee'…<br>(Fieldnote, Willow, 03/01/2018, DS)<br><br>Mo hands a prescription over to Rashida part way through its preparation asking her to *'take over please'* - a quiet negotiation met with no resistance and a seamless transfer of a task part way through. Mo says to Ali, who's recently arrived for his shift, *'we need to check the orders'*. Ali lifts up a heavy crate of medicines from the floor and they both go to the back of the pharmacy.<br>(Fieldnote, Willow, 03/01/2018, DS) |

Continued

**Table 4** Continued

| Routines involving the printed copy of the electronic prescription | Examples from fieldnotes and interviews |
|---|---|
| Prescriptions filed for collection act as a record to check the right patient has the right drug | All four pharmacies implemented similar process for patients to pick up their medicines. This description from Foxglove denotes how it's done:<br><br>After checking the medicines in the basket with the medicines listed on the prescription, Zane, the pharmacist bags up the medicines and seals the bag with a pharmacy bag label. The prescription is put in a white plastic basket that will go through to the front counter for filing alphabetically by patient's surname in a drawer under the counter. When a patient comes in to pick up a prescription the staff search through this filing system for the prescription. Aiza tells me the most important thing to check is the patient's address: *'so many patients here have the same name.'* Sameer, at Willow also echoed this point: there are so many people with the same name at the pharmacy that staff *'can't go on name. I don't care if (a patient) can't speak English. If you can't say your address you won't get your prescription.'* He draws my attention to the reminder notes stuck to the collection drawers instructing staff to search on address first and to match the bagged up drugs to the prescription.<br>(Fieldnote, Foxglove and Willow, 10/01/2019, 16/01/2019 & 24/01/19, NF) |
| The prescription becomes part of a multi-faceted checking process, alongside a master document and the dosette box itself | Anjali, a pharmacist, takes me through the dosette checking process. She starts with the MAP—a paper record of which drugs the patient is on and at which time points during the day they should be taken. She reiterates that *'if any changes happen, changes must be noted down on the MAP by hand.'* The next thing is to check the prescription with the MAP, but 'the MAP is gospel'. Anjali explains the next step is to 'go down the MAP and tick off on the prescription if the drugs are on the MAP. It's also a double check on the labelling process.' While I'm doing this, says Anjali, I also check on the doses etc. Once these checks have been done, Anjali puts the prescription away in the basket that belongs to the patient whose dosette she is checking, before she can begin to check the capsules and tablets in the dosette box.<br>(Fieldnote, Lilac, 14/03/2019, NF)<br><br>There were two people working in the dosette area on my arrival—Saleem, the second pharmacist and Nadira, an apprentice technician. Saleem inspects a dosette, flicking his finger underneath to make the tablets jump and rotating it at various angles to inspect inside the individual cells. He then signs line by line against each medication on the printed lists stuck into the dosette box cover. This one had *'no complications.'* The dosette area is a narrow rectangular room with counter down the left hand side as you enter from the dispensary. Saleem is on the right hand side of the room, standing up and perched behind a stack of green crates all containing dosettes that need checking. Saleem tells me that he first checks the prescription against the *'card'* (a plastic wallet containing a printed Word document with the patient's name, medicines, medication review date). This is also stored in electronic form on the computer. Having established that the card and the prescription are the same, he then checks the contents of the dosette box. He explains that the dosette box, the card and the prescription need to agree with each other. My sense is that *'the card'* is the working or master document in guiding their practices around checking and making up the boxes.<br>(Fieldnote, Foxglove, 05/11/2018 DS) |

Caleb who has been in '*production*' appears and strikes up a conversation with Jimi, the pharmacy owner—a discussion about Epilim [sodium valproate] 100 and 200 and stability issues. Caleb suggests that as they use so much of these items in one production run he wonders if they could change its status from '*external*' to '*production*'. Jimi considers it and answers back '*if we only filled it once at production, turnover is high.*' Mo, the pharmacist comes over and joins in, suggesting they ask someone. Jimi suggests that Caleb phone Sanofi, the drug company, to ask how long the tablets can remain out of the packets.

(Fieldnote 09/01/2018, Willow pharmacy, DS)

'Externals' add considerable extra work to dosette production as they are dealt with manually, but this exchange also highlights how notions of 'safety' incorporate understandings of supply and demand, drug-related storage and stability, alongside a welcoming of ideas by senior staff.

We also observed open discussion as challenges arise, with junior members of staff taking the initiative to come up with solutions or refine working routines based on their knowledge and expertise:

Jimi addressed a small group of staff, reflecting on a recent '*deblistering problem*' when more than one brand of pill ended up in the same container. He told them to '*be alert*' and made some suggestions for avoiding this in future, acknowledging how difficult it is amidst a '*sea of gliclazide*'.

A few days later, Laila (dispenser responsible for dosettes) spends her lunch break creating a '*deblistering SOP*'. With three steps typed up, she asks Saleem, a pharmacist, what step four should be, adding '*I can't believe this—it is like the easiest thing we do*' By the end of the lunch break she has five steps: '*I will show it to Sameer and Caleb…this is what we do*'.

I was struck by the bottom-up, collaborative nature of this exercise. Safety was being worked out on the hoof with everyone included, acknowledgement of its complexity and overall, a sense of commitment and fun. But the formalisation of this into a document seemed at odds with their usual approach of just asking each other.

I learn a few days later that the SOP is now an A4 sheet with 12 bullet points stuck on the wall of which Laila is clearly proud. Nina watched Laila stage a quiz in which she tested Linda's knowledge, with Naihra joking from the sidelines. It is imperfect in its detail, and appears incomplete.

Nina asked '*Why did you have to do a SOP…have you devised a new routine?*'

Laila '*No, it was in our heads. Everyone knew, but it wasn't written down*'

Everyone was happy. The equilibrium was restored. The SOP is not so much an instruction of what to

do but a reflection of what they now do, a product of teamwork and a consolidation of their collective knowledge.

(Fieldnotes, Willow, 23/01/2018, DS)

This care for one another created an environment which allowed staff to work collaboratively and openly resolve errors. At Willow and Foxglove in particular, staff were encouraged to identify and discuss errors which were regarded as an inevitable aspect of best practice:

if you haven't spotted any today, are you doing your job properly or are you asleep on the job, just letting things through? Errors keep you on your toes.

(Interview, Pharmacist, Foxglove, 04/02/2019)

In Woodland pharmacies, the category of 'error' was noticeably broad and included: discrepancies arising when patient's medication changed on hospital discharge; robot errors (eg, broken tablets; tablets 'jumping' between cells). One consequence of this broad description that incorporated both 'human' and 'technological' error was that it made talk about errors commonplace, easy, collegiate and—importantly—actionable:

Aiza has grabbed one set of prescriptions belonging to one patient, or so it would seem. Quickly she has spotted an error and tells me '*this is something you should see. There's a mix up on names. Two patients with same surname but different addresses—someone has put both prescriptions together.*' Aiza lays out the four prescriptions on the island worktop for me to see that indeed two belong to a 'D' and two belong to a 'S' but they have the same surname. She shows Raheem the error—'*I almost made up one prescription for two people!*' She does this in a matter of fact, upbeat way. There's no blame that it was someone else who put the prescriptions together initially. It's as if she's taken responsibility for almost making the error herself, had she not checked the names on the top of the prescription so carefully.

(Fieldnote, Foxglove, 16/01/19, NF)

### Caring for the patient

Finally, safety emerges out a shared concern to care for patients; automation has not diminished relationships between staff and patients ('*we know our dosette patients*'). There are sticky notes and '*handover sheets*' on the dispensary walls acting as reminders of: '*dosette patients*' holidays/ extra dosettes needed; hospital admissions; medication changes. Staff often made themselves available for dosette queries on their days off.

At Poppy, Marie, a dispenser, explained how she used her initiative to adapt their routine for labelling dosettes to accommodate a patient with too many medicines for one dosette (22 morning; 17 midday; 6 teatime; 18 bedtime). Usually one inlay is placed in each dosette listing the medication it contains, its appearance and dose instructions. Marie explained '*we actually make two dosette boxes for her, because it doesn't all fit into [one] box. So she has

*her pain medications in one box, and in the other box she has the rest.*' When Marie imagined how the dosette might be used beyond the pharmacy, she placed two inlays within each dosette, so each box listed the patient's full regimen:

> So if she ever went into hospital, say if she only grabbed one dosette box and she only took the pain one and they thought, oh, she's only on painkillers, and she didn't take the others (…) It will still have the whole list, so then they'll say to her, 'Oh, where's your other medication because it's on this?' (…) Yeah, it dawned on me and she has to go back into hospital, and I don't know what it's like, if she's elderly and she's rushing, she might only pick up one dosette, and then what? The hospital aren't going to know what she's on.
>
> (Interview, Dispenser, Poppy, 29/09/2019)

The following extract expresses the reciprocity and strength of staff-patient relationships which endure and surpass the role of automation:

> Laila and Saleem are back from their surgery visits to pick up the daily prescriptions. Laila is full of chatter and excitement and has a story to tell. She says that she went into the surgery and it was so full, people everywhere and a really long queue. As she waited, a man, '*whose dosette we make up*' came up to her and '*touched her feet, thanked her and said pray for me.*' The three of us talk about what this might mean. I asked Laila '*is he grateful because you make up his dosettes?*' '*I don't know*' she replied, '*but I was so embarrassed. Everyone was watching me.*' Saleem asked if she was '*anything to him, like an aunty?*' 'I'm nothing to him!' exclaimed Laila, '*I'm just the girl in the pharmacy.*'
>
> (Fieldnote, Willow, 08/01/2018, NF)

Several participants emphasised the importance of getting the right drug to the right patient at the right time as their top safety priority. Staff speculated on imaginary patients and future imagined scenarios by way of performing safety here-and-now, maintaining a rhetorically persuasive account of their high risk environment and embedding these future abstractions into material structures and systems in-house[35]:

> You have to be accurate on what you do. Be prepared, because obviously you have to always double check what you do. It doesn't matter whichever job you do, and especially if you work in a pharmacy with medication, it's got to do with someone's life. As I said, if you give them something wrong, they could end up in hospital, they don't know where they're going to end up; they might die, or whatever!
>
> (Interview, Dispenser, Foxglove, 04/02/2019)

> Yes, my main worry is a mistake being made, so I have that at the top of my brain all the time. I'm constantly checking that everything is right, that it is the right meds, it's the right strength, you know the

right tablet, every step. (…) Because I'd hate it. (…) like if it was my nan's meds or something like that and she said, "Oh, a mistake has been made and I've taken the wrong meds!" I wouldn't be very happy, I'd think, well, I think the whole point of the dosette box is to make sure that vulnerable people, like elderly or those who can't handle their meds, it's done for them in the correct way. (…) It is a constant worry, it's a lot of pressure.

(Interview, Dispenser, Lilac, 13/01/2020)

### Polypharmacy as a safety issue: whose responsibility?

Our study revealed an important paradox. On the one hand, polypharmacy was visible, pervasive and productive of numerous working routines. On the other, polypharmacy per se was rarely discussed as a safety concern, either between pharmacy staff or between pharmacist and GP. Safety meant close attention to practices ensuring medicines were dispensed as prescribed, and correcting errors pertaining to individual drugs through the clinical check. It did not mean actively challenging polypharmacy per se, even in situations where the volume of prescribing (10+ or 15+ items) might indicate 'high risk'.

Table 5 illustrates the tension pharmacy staff articulate between dispensing 'safely' in the context of inherently risky lists of multiple medicines, from a professional position which distances them from the act of prescribing, the responsibility for which is firmly with the prescriber, usually the GP.

### DISCUSSION

Our analysis illuminates the ongoing, collaborative work that constitutes safety across four community pharmacies. The end towards which this work is focused is 'safe' dispensing of medicines, broadly understood as 'right drug, right patient, right time'. The means by which this is achieved is a highly nuanced, ongoing process of organising and reorganising, negotiation and renegotiation. Safety is not fixed or inflexible—as might be assumed in a system standardised by protocols—but is in constant flux and open for adaptation by staff at all levels. From this perspective safety is a verb, not a noun.

Patient safety was not assured *because of* the implementation of technologies such as dispensing and dosette robots, dosettes or electronic prescribing, but emerged out of a shared concern by pharmacy staff to 'care'. SOPs did not drive action, but emerged out of collective action; the most useful ones were unfinished 'work-in-progress', flexible scripts that remained open to further adaptation (eg, the deblistering SOP, the robot 'protocol'). In addition to learning from errors, staff shared stories of caring for a collective imaginary—imagined scenarios of what might happen if medicines are not dispensed carefully and safely. The collective imaginary is rhetorically powerful, effective in sustaining staff orientation towards safety practices in an inherently risky context. Care is the glue that ensures patient safety and encompasses care for

| Table 5 Polypharmacy and safety | |
|---|---|
| **Tension pharmacy staff face between dispensing multiple medicines safely and ability to challenge instances of polypharmacy** | **Examples from fieldnotes and interviews** |
| Clinical checks and the distancing of the pharmacy from polypharmacy as a safety concern | When I ask Mo about checking the dosettes, he says this is a *'clinical check and an accuracy check'*. When I ask what a clinical check is he tells me it is about looking for things like interactions between the meds and checking if a dose is too high for example. He goes on to explain that it might be *'accurate but not safe!'* Mo says that the checks will have been done in the last round (by which he means same patient, last month's round of dosettes) but that they check every round nevertheless. I cautiously ask about the fact that some patients are on very long lists of medicines—as this is something we are particularly interested in. He says that often there is a *'primary'* condition but that sometimes drugs cause side effects and this might lead to other prescriptions. Mo concedes this is difficult, and goes on to explain that the (pharmacy conducted) Medicine Use Reviews are mainly about how and if the patient is using the meds… if there are adherence or side effects problems—but it's not a *'full clinical'* review about whether the drugs are working for example. He says that this is mainly the responsibility of the prescriber. (Fieldnote Willow, 11/12/2018, DS)<br><br>NF: Is it ever the pharmacist's role to talk to patients about stopping the medicines or …?<br>Zane: Sometimes … well, we wouldn't initiate the stop here, no.<br>NF: Right.<br>Zane: If there's a stop, like for example, if a patient has been given an antibiotic, some of them interact with cholesterol medications, so you just need to let them know to stop taking that (cholesterol) medication for a week.<br>NF: Like temporarily?<br>Zane: Yeah, but it's just a temporary stop. We never say completely stop taking it, because ultimately that's the doctor's decision. (Interview, Pharmacist, Foxglove, 06/02/2019) |
| Polypharmacy as a norm but staff not in a position to challenge it | In this interview, Raheem recounted his shock of having to dispense a prescription of over 10 items of medication to a young child. NF asked him if he had the same sense of shock when he's making up a prescription of many items for an older person:<br>Raheem: Yes, but it's not as much because—I know it sounds bad—but I'm a bit used to it here. And I seem more shocked when I see a 65 year old with less medication than usual.<br>NF: Oh right<br>Raheem: We actually had a lady that was around about 66 or something, she only took two medication and that just for pain maybe, and it was no diabetes, gastric or anything else, so I was shocked. (Interview, Technician, Foxglove, 04/02/2019) |

relevant technologies, care for each other and care for the patient. Following a Safety II approach, we have shown how pharmacy staff continuously adapt their routines to ensure 'as many things as possible go right' and that medicine safety can be assured 'to succeed under varying conditions'.[19] Sophisticated understandings of how the everyday actions of healthcare staff produce safety are essential as the organisational contexts within which healthcare is delivered become increasingly complex.

The absence of explicit talk about polypharmacy, or the relationships between polypharmacy and safety—even though we became known as the 'polypharmacy researchers'—surprised us. This was not unique to the pharmacies in our study; we did not observe naturally occurring talk about polypharmacy in the GP practices taking part in our wider APOLLO-MM study either (not yet published). While the pharmacists, technicians and dispensers worked hard to ensure 'safety' for their patients

affected by polypharmacy, they did not consider they had any legitimate warrant to challenge it, and located the responsibility for the prevalence of polypharmacy with different parts of the health system, usually GPs. The MUR—as the name suggests—was framed by participants as an exploration of 'use' of drugs, not an opportunity to question polypharmacy per se. Patients affected by polypharmacy did not constitute a target group for MURs.[36]

### Strengths and weaknesses of this study
To our knowledge, this is the first ethnographic study conducted in community pharmacy that has focused on polypharmacy and its intersection with safety practices. A key strength of our ethnographic approach was the opportunity to spend many hours observing the detailed practices of staff 'doing safety' in dispensing medicines in the particular context in which it happens, rather than relying on abstracted interview accounts alone. Although

our pharmacies varied in terms of the populations they served, all belonged to independent pharmacy groups (3–5 pharmacies in each group). Our findings may not translate readily to larger chain pharmacies, but our interpretations may offer useful ways of conceptualising safety across UK and international settings.

Not everything was easily visible to us as ethnographers. We did not have access to some aspects of pharmacy work, such as management decisions about 'running the business' of a community pharmacy, or the relationship between the financial and clinical dimensions of pharmacy work. This may have special relevance in polypharmacy in a health system such as the NHS where dispensing fees are paid to pharmacies on a 'per item' basis. Our study contributes to a body of qualitative research by foregrounding 'hidden work' and illuminating the creative 'tinkering', practical judgements and situated knowledge that is often missing from professional accounts and policy documents, but which is essential to ensuring technology assisted routines are safely implemented.[13 37–40] In contrast to previous work conducted in pharmacy settings our observations show that staff are adept at maintaining safe practices and resolving errors despite constraints such as limited space and interrupted work flows.[41 42]

## The meaning of the study

Our study sheds light on the often hidden work that pharmacy staff undertake in increasingly complex, high-risk settings fueled by escalating prescribing. While some policy literature acknowledges a need to help 'practitioners manage workload related to polypharmacy in order to improve medication safety',[6] we argue that policy-makers could take greater account of practice 'on the ground' to inform guidelines by acknowledging how pharmacy professionals are currently working to ensure medication safety in high risk situations, such as those presented by polypharmacy. Furthermore, the policy literature focuses on distinguishing between appropriate and problematic polypharmacy.[18 43 44] However, these are terms that we did not hear used in the pharmacy settings we were given access to. This suggests that policy level pronouncements have not found their way to those on the ground or that the terms used by policy makers and academics do not resonate with those working on the front line of polypharmacy.

## Future research

There have been calls for a greater role of pharmacists in managing high risk and problematic polypharmacy,[6 45] including the integration of clinical pharmacists within primary care to deliver Structured Medication Reviews under the new 2020/21 GP Contract to patients affected by complex problematic polypharmacy.[46] Community pharmacists in our study did not feel it was within their remit to challenge prescribing regimens initiated by other healthcare professionals in the healthcare system. Further research is needed to understand how these different professionals (community pharmacists; clinical pharmacists within general practice settings; prescribing doctors and nurses) and new professional arrangements work together to negotiate and address complex polypharmacy and with what consequences for patients. We will consider the role of GPs and other health professionals in supporting or challenging polypharmacy in our wider research.

**Acknowledgements** We would like to thank staff from the four community pharmacies who agreed to participate in this study, colleagues at QMUL who commented on an early draft of the manuscript, and our Expert Advisory Group and Patient Panel for their support of our research.

**Contributors** Both authors meet the ICMJE criteria for authorship. The study was conceived by DS. Data gathering and data analysis were jointly undertaken by DS and NF. NF wrote the first draft of the paper. Both authors revised and finalised the manuscript.

**Funding** This work was supported by the National Institute for Health Research (NIHR) Clinician Scientist Award number CS-2015-15-004. Additionally, this work was supported by the NIHR Collaboration for Leadership in Applied Health Research and Care (CLAHRC) North Thames.

**Disclaimer** The views expressed are those of the author(s) and not necessarily those of the NHS, the NIHR or the Department of Health and Social Care.

**Competing interests** None declared.

**Patient consent for publication** Not required.

**Ethics approval** The project has ethics approval from Leeds West Research Ethics Committee (IRAS project ID: 205517; REC reference 16/YH/0462).

**Provenance and peer review** Not commissioned; externally peer reviewed.

**Data availability statement** No data are available. Our ethics approval and consent procedures were based on the anonymity of the individuals who participated, hence further access to the full data set cannot be granted.

**ORCID iDs**
Nina Fudge http://orcid.org/0000-0002-7161-4355
Deborah Swinglehurst http://orcid.org/0000-0003-1261-9268

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
