## [Reviewer comments · BMJ Open]

ARTICLE DETAILS

TITLE (PROVISIONAL)	"It's all about patient safety": an ethnographic study of how pharmacy staff construct medicines safety in the context of polypharmacy."
AUTHORS	Fudge, Nina; Swinglehurst, Deborah

VERSION 1 – REVIEW

REVIEWER	Rachel Savage and Paula Rochon Women's College Research Institute, Women's College Hospital, Toronto, ON CANADA
REVIEW RETURNED	06-Oct-2020

GENERAL COMMENTS	This is an ethnographic study of how 'safety' is constructed and accomplished within four English community pharmacies. The work is oriented with a "Safety-II" view and wishes to examine safety specifically in the context of polypharmacy. Polypharmacy is an important public health issue and this work adds a unique perspective to the literature on this topic. The analysis was consistent with the described orientation and the data presented nicely illuminated the story authors wished to tell. Description, analysis and interpretation were well-balanced in this paper. Major comment: Our major or substantive comment is that more work is needed to integrate and situate the findings of this study in the broader literature of medication safety and pharmacist roles. Your finding re: the absence of talk of polypharmacy, or the terms inappropriate vs. appropriate, is particularly interesting and undertheorized in your discussion. Triangulation with other literature would serve to better understand what can be inferred and learned from your observations. Placing your findings in the context of the broader literature may help provide more concrete or tangible suggestions for future research, as well as community pharmacy practices and policies. Other minor comments: In general, this paper would benefit from additional editing to make it more concise. At present it exceeds the recommended word count. Introduction 1. Lines 3-5, p 6 - Could the authors please elaborate on the "Safety-II" view to explain what is gained by observing where practice occurs error-free (vs. the traditional Safety-I orientation)? This is important to understanding the appropriateness and value of this orientation for your work and how it may contribute to new ways of understanding this important topic.
---

	Methods 1. Sampling - Line 7, p 6 – How many community pharmacies were invited to participate and were there any refusals? How were staff selected for interview? Eligibility criteria? 2. Table 1 – In the context of polypharmacy, it would also be helpful (beyond deprivation) to understand more about the demographic characteristics of the residents. For example, the percentage of residents in the neighbourhoods (for example those aged 65 years and older; women and men) 3. Table 2 – It seems there may be other important characteristics of interviewees to better understand whose perspectives are being represented by this work. Sex, level of experience/training, for example. Is there additional information you can provide? 4. I would think it would be important to include the ethics statement in the text of the methods section, rather than at the end of the paper. Results 1. Relative to other findings, the data presented on the robot (caring for technology) seemed lacking in richer detail and from what was presented, it was unclear how it enhanced medication safety, other than staff simply saying it did. Was there anything unique about how staff interacted with the robot that led to medication dispensing that was more careful or thoughtful in some way? If yes, this would be useful to present and would align better with the orientation of your paper. If no, this should be made more explicit, as that is an interesting finding in and of itself. 2. Line 42, p 12 – Presenting claims from the robot manufacturer’s website seemed out of place, as it was not a “document” that you identified in your methods section. It took me some time to realize that its inclusion and analysis was your (as the analyst) interpretation of the results, an attempt to explain where staff narratives on the robot may be constructed from. This should be made clearer – it is important to clearly convey when you move from your research participants’ accounts or observations to your own interpretations (e.g. delineating results vs. your interpretation as you did in other sections by using “we interpret”). Discussion 1. Lines 7-9, p 26 and Lines 16-18, p 27 – Authors should cite the work they refer to (“This was not unique to pharmacies; we did not witness naturally occurring talk about polypharmacy in our study GP practices either.”; “the policy literature focuses on distinguishing between appropriate and inappropriate polypharmacy.”)
--	--

REVIEWER	Ayesha Siddiqua King Khalid University, Saudi Arabia
REVIEW RETURNED	08-Oct-2020

GENERAL COMMENTS	I would like to thank the authors to read their manuscript. Please consider changing the way the references are presented. It would be better to check the journal’s norms and condition of presenting the references.
--

VERSION 1 – AUTHOR RESPONSE

Reviewer 1: Rachel Savage and Paula Rochon	
This is an ethnographic study of how 'safety' is constructed and accomplished within four English community pharmacies. The work is oriented with a "Safety-II" view and wishes to examine safety specifically in the context of polypharmacy. Polypharmacy is an important public health issue and this work adds a unique perspective to the literature on this topic. The analysis was consistent with the described orientation and the data presented nicely illuminated the story authors wished to tell. Description, analysis and interpretation were well-balanced in this paper.	We are very grateful to the reviewers for reading our paper and providing such positive and constructive comments.
Major comment: Our major or substantive comment is that more work is needed to integrate and situate the findings of this study in the broader literature of medication safety and pharmacist roles. Your finding re: the absence of talk of polypharmacy, or the terms inappropriate vs. appropriate, is particularly interesting and undertheorized in your discussion. Triangulation with other literature would serve to better understand what can be inferred and learned from your	We thank the reviewers for this comment. On this suggestion we have revisited the broader literature of medication safety and pharmacist roles. This is a huge literature and much of it is beyond the scope of our current study. Of note we have found only five published ethnographic studies conducted in community pharmacy settings and of two were explicitly concerned with safety, but none focused on the intersection between safety practices and polypharmacy as a particular focus of interest. We have drawn attention to this in the manuscript. We have also extended the final section of our Discussion to acknowledge the recent publication of the Service Specification for structured medication reviews under the new GP contract which was published since our first submission. In particular we highlight the challenge that these new arrangements of professionals may create given our finding that community pharmacists did not feel well placed to challenge prescribing decisions.

observations. Placing your findings in the context of the broader literature may help provide more concrete or tangible suggestions for future research, as well as community pharmacy practices and policies.	
Other minor comments: In general, this paper would benefit from additional editing to make it more concise. At present it exceeds the recommended word count.	We are aware that our paper currently exceeds the word count though we also note that the word count is flexible. It is a challenge to produce the ‘thick descriptions’ (see above) that are required of a good ethnography within the same word limits allowed for a quantitative paper. We are concerned that reducing the word limit will reduce the paper’s clarity for the reader but we have reduced the word count where possible (see marked version of the manuscript).
Introduction 1. Lines 3-5, p 6 - Could the authors please elaborate on the “Safety-II” view to explain what is gained by observing where practice occurs error-free (vs. the traditional Safety-I orientation)? This is important to understanding the appropriateness and value of this orientation for your work and how it may contribute to new ways of understanding this important topic.	We have reworded this sentence and hope that this more clearly explains the value of focussing on ‘ordinary’ every day practice to learn about safety in complex environments, rather than focussing on rare incidents. The sentence now reads (page 4-5, marked version): A more nuanced approach is now emerging, shifting the focus away from how errors are produced and avoided (which draws attention to a minority of incidents) towards what can be learned from observing ordinary everyday performance, where – mostly – practice occurs error-free (a ‘Safety II’ approach).¹⁹
Methods 1. Sampling - Line 7, p 6 – How many community pharmacies were invited to participate and were there any refusals? How were staff selected for interview? Eligibility criteria?	The sampling and recruitment methods of the study are explained in detail in our protocol paper (https://bmjopen.bmj.com/content/9/8/e031601). We started by recruiting GP practices who then indicated pharmacies they worked closely with and which a substantial number of their patient population used, who may be interested in participating in the research. Thus four pharmacies were invited to take part (suggested by three GP practices) and all four accepted the invitation to take part. We ensured that our sample pharmacies and GP practices were from contrasting urban and suburban areas. We have referenced the protocol paper which explains the methods in more detail. We are reluctant to amend the text as this will add to the word count. We have added some further explanation of how we selected staff for interview, essentially staff who we had observed and shadowed during participant observation who we knew to be

	involved in work relevant to polypharmacy and its safety. Eligibility criteria was that they worked in the pharmacy. The amended text now reads (marked version, page 9): We conducted 19 formal interviews with 21 pharmacy staff identified through ethnographic observations as doing work relevant to polypharmacy and its safety (including one group interview) (see table 2).
Methods 2. Table 1 – In the context of polypharmacy, it would also be helpful (beyond deprivation) to understand more about the demographic characteristics of the residents. For example, the percentage of residents in the neighbourhoods (for example those aged 65 years and older; women and men)	We have added some further detail about the setting where the pharmacies are located. We have provided % of the population who are 65 years and over and of this older population, % who are male and female (see table 1, column 1). We have reordered columns 2 and 3 so the table is more readable.
Methods 3. Table 2 – It seems there may be other important characteristics of interviewees to better understand whose perspectives are being represented by this work. Sex, level of experience/training, for example. Is there additional information you can provide?	We have added the numbers of participants who are male and female (see table 2, columns 2-6). In terms of training/experience the job descriptor headings give the reader a sense of position of responsibility and level of training. We have kept the table as is in this respect so as not to add to the word count.
Methods 4. I would think it would be important to include the ethics statement in the text of the methods section, rather than at the end of the paper.	We followed BMJ Author Hub guidance for supplying a statement about ethics approval (along with other statements such as funding, competing interests etc) and these all appear at the end of the paper (based on guidance and looking at recently published original research). We would be happy to insert the ethics statement into the methods section should the editors state that this is editorial policy.
Results 1. Relative to other findings, the data presented on the robot (caring for technology) seemed lacking in richer detail and from what was presented, it was unclear how it enhanced	The purpose of our study cannot conclude whether producing dosettes by robot rather than by hand produces safer dispensing. We can only comment on the staffs' narrative justifying the robot that it leads to safer dispensing, mirroring the manufacturer's claim. However, we are making the point that simply introducing automated technology for dispensing does not result in safety - indeed staff are required to incorporate new processes and procedures to ensure medicines are dispensed into dosette boxes as safely as

medication safety, other than staff simply saying it did. Was there anything unique about how staff interacted with the robot that led to medication dispensing that was more careful or thoughtful in some way? If yes, this would be useful to present and would align better with the orientation of your paper. If no, this should be made more explicit, as that is an interesting finding in and of itself.	possible. It creates new work and new potentials for error and the humans must manage and anticipate this. We would have loved to include more detail on the robot but were restricted by word count. The robot was a feature in only one of the four pharmacies whereas EPS featured in all four pharmacies - another reason why the robot technology may seem less detailed and rich compared to the results on EPS as a technology.
Results 2. Line 42, p 12 – Presenting claims from the robot manufacturer’s website seemed out of place, as it was not a “document” that you identified in your methods section. It took me some time to realize that its inclusion and analysis was your (as the analyst) interpretation of the results, an attempt to explain where staff narratives on the robot may be constructed from. This should be made clearer – it is important to clearly convey when you move from your research participants’ accounts or observations to your own interpretations (e.g. delineating results vs. your interpretation as you did in other sections by using “we interpret”).	We listed types of documents in the methods as examples but this list wasn’t meant to be exhaustive. For clarity we have added another category of document to refer to technology/manufacturer’s guidance See marked version, page 8: documents (e.g. standard operating procedures, dosette checklists, to do lists, manufacturer’s guidance identified as relevant through our observations and interviews). And we have made the link between what participant’s said and statements in the manufacturer’s website: See marked version, page 14: This well-rehearsed collective narrative appealed to staff and drove the implementation and ongoing use of the robot, although staff were never explicit about what constitutes ‘safety’ nor how the robot contributed to it. We interpret staffs’ statements about safety as resonating with statements from the robot manufacturer, such as ‘increased accuracy’ compared to the ‘manual preparation method’ enabling ‘the pharmacy to greatly increase safety’ (Document: robot manufacturer’s website).
Discussion 1. Lines 7-9, p 26 and Lines 16-18, p 27 – Authors should cite the work they refer to (“This was not unique to pharmacies; we did not witness naturally occurring	The work we were referring to in lines 7-9 was observations we made in the GP practice sites of our broader research study, although we have not published this work yet. We have clarified this by amending the sentence to: See marked version, page 29: This was not unique to the pharmacies in our study; we did not observe naturally occurring talk about polypharmacy

talk about polypharmacy in our study GP practices either.”; “the policy literature focuses on distinguishing between appropriate and inappropriate polypharmacy.”)	in the GP practices taking part in our wider APOLLO-MM study either (not yet published). We have provided references for the policy literature on appropriate and problematic polypharmacy (see marked version, page 31). We are referring to ‘inappropriate polypharmacy’ as ‘problematic polypharmacy’ although we recognise the terms used are interchangeable.
Reviewer 2: Ayesha Siddiqua	
I would like to thank the authors to read their manuscript.	We are very grateful to the reviewer for reading our paper.
Please consider changing the way the references are presented. It would be better to check the journal’s norms and condition of presenting the references.	We have amended references according to the journal requirements and removed two references in author date format which had been inadvertently left in.